# A Stepwise and Hybrid Trust Evaluation Scheme for Tactical Wireless Sensor Networks [note 1]

**DOI:** 10.3390/s20041108

**Published:** 2020-02-18

**Authors:** Jihun Lim, Dooho Keum, Young-Bae Ko

**Affiliations:** Department of Computer Engineering, Ajou University, Suwon 16499, Korea; limbee94@ajou.ac.kr (J.L.); dooho1000@ajou.ac.kr (D.K.)

**Keywords:** wireless sensor networks, tactical networks, trust management

## Abstract

In tactical wireless sensor networks, tactical sensors are increasingly expected to be exploited for information collection in battlefields or dangerous areas on behalf of soldiers. The main function of these networks is to use sensors to measure radiation, nuclear, and biochemical values for the safety of allies and also to monitor and carry out reconnaissance of enemies. These tactical sensors require a network traffic flow that sends various types of measured information to the gateway, which needs high reliability. To ensure reliability, it must be able to detect malicious nodes that perform packet-dropping attacks to disrupt the network traffic flow, and energy-constrained sensors require energy-efficient methods to detect them. Therefore, in this paper, we propose a stepwise and hybrid trust evaluation scheme for locating malicious nodes that perform packet-dropping attacks in a tree-based network. Sensors send a query to the gateway by observing the traffic patterns of their child nodes. Moreover, depending on the situation, the gateway detects malicious nodes by choosing between gateway-assisted trust evaluation and gateway-independent trust evaluation. We implemented and evaluated the proposed scheme with the OPNET simulator, and the results showed that a higher packet delivery ratio can be achieved with significantly lower energy consumption.

## 1. Introduction

In tactical wireless sensor networking environments, various tactical sensors are deployed to collect information from battlefields or dangerous areas on behalf of soldiers in consideration of their safety. An unattended ground sensor (UGS) is one of the tactical sensors used for this purpose. A UGS is a long-range wireless communication sensor that can be installed and retrieved directly. It is divided into two types: tactical-UGS is a compact ground-based sensor deployed on the battlefield that collects chemical, biological, radiological, and nuclear information, and urban-UGS, a sensor that is installed in urban areas for surveillance of enclosed areas such as buildings, sewers, and tunnels. These sensors have gateways to collect data and send them to an external command center. The command center will be able to analyze and monitor the battlefield situation and conduct tactical command using the large amount of data received from the UGS. The UGS and the gateway periodically share and update information on the battlefield situation. On the battlefield, the UGSs primarily communicate in a star topology, in which sensor nodes are connected directly to the gateway in one hop. However, in long-range wireless sensor networks configured with a star topology, sensors communicate at very low data rates of 250 kbps [1]. This may be undesirable to the sending of the aforementioned tactical data information (surveillance video, chemical, biological, radiological, and nuclear). According to a UGS report, the requirement for long-range communication is 3 km, whereas in the test, one-hop communication maintains the connection up to 800 m [2]. To overcome the range problem of one-hop communication, UGS must communicate in a tree-based sensor to the gateway multi-hop networks. Figure 1 shows a scenario of UGS communications, where the UGS collects information (chemical, biological, radiological, nuclear, and surveillance video) and delivers it to the gateway via multi-hop communication.

Moreover, tactical wireless sensor networks are vulnerable to nodes performing malicious attacks. Especially, because of continuous packet-dropping attacks, which prevent the transmission of important data, it is difficult to gather information about the battlefield situation. Common malicious attacks include packet-dropping attacks (e.g., black-hole), selective forwarding attacks (e.g., gray-hole and on–off), and wormhole attacks [3]. Among them, selective forwarding attacks, which drop packets even while maintaining a certain level of trust, are considered seriously intelligent malicious attacks in multi-hop tactical networks [4]. This type of attack is difficult to mitigate because the node changes the attack period depending on the situation. Preventing such attacks requires a trust evaluation technique that can detect a malicious node using its reputation. There are two main types of trust evaluation techniques, which are summarized in the Related Works section: The first type is the gateway-assisted trust evaluation approach, in which the gateway or cluster head with a higher computing power than that of other nodes performs a trust evaluation of all child nodes [5,6,7]. The second one is the gateway-independent trust evaluation approach, in which all nodes perform a trust evaluation of their surrounding nodes [8,9,10,11].

The existing solutions have a common problem of high overhead because every node performs a trust evaluation periodically regardless of the presence of an attack. Because every operation needs to be energy efficient in this resource-constrained network, we use a stepwise trust evaluation framework that can detect malicious nodes efficiently in tree-based networks [12]. In this framework, not all nodes participate in the trust evaluation and typically observe suspicious behavior only in their own child nodes. When a node observes suspicious behavior from child nodes, it sends a query to the gateway. On the basis of the received query, the gateway instructs the nodes around the suspicious node to perform a trust evaluation. The gateway detects malicious nodes with a hybrid application of gateway-assisted and gateway-independent trust evaluations. Moreover, malicious nodes can be detected by appropriately changing the malicious node threshold according to the reliability of each node. From the result of this stepwise trust evaluation, we conclude that we can reduce the energy consumption used for the trust evaluations.

The contributions of the proposed scheme are threefold: First, as the trust evaluation is carried out in a stepwise manner, the network can be more efficient in terms of overhead and energy. Second, nodes in the parent-only evaluation region can prevent false positives as the trust evaluation is made through drone nodes that perform a physical inspection. Finally, we can detect attacks using an adaptive detection threshold as the situation changes.

Before going into the specific details of the protocol, we will discuss the related works in Section 2. In Section 3, we present our proposed scheme. The performance evaluation is discussed in Section 4. Finally, we conclude our paper and discuss the summary in Section 5.

## 2. Related Works

In this section, we introduce related works whose proposed schemes use one of these two approaches: gateway-assisted trust evaluation and gateway-independent trust evaluation.

### 2.1. Gateway-Assisted Trust Evaluation

Tajeddine et al. [5] proposed a routing protocol named Centralized Trust-based Efficient Routing with Authentication (CENTERA), which uses a gateway-assisted trust evaluation technique. CENTERA utilizes a powerful base station (BS) to gather trust information from all nodes in the topology and calculates the best routes after detecting and isolating different types of malicious nodes. The BS creates a global view of the network topology and calculates three metrics, namely, maliciousness, cooperation, and competence, for the evaluation of the trust value of each node. The BS can detect types of malicious nodes, such as a malicious node sending false or illogical information, a non-cooperative node not reliably forwarding packets of other nodes, or an ineffectual node unable to correctly deliver packets to the BS. Malicious nodes are isolated for a certain period time depending on their history. BS increments the bad or probation level of every node with bad behavior, whereas decrements this level for good behaviors. The BS uses an efficient method to periodically disseminate the updated information to all nodes.

Bao et al. [6] proposed a trust management protocol for wireless sensor networks that uses a gateway-assisted trust evaluation technique. This protocol utilizes two levels of hierarchy to manage trust. The sensor node layer reports the results of the peer-to-peer trust evaluations of other sensor nodes to the cluster head node. The cluster head node performs a cluster head-sensor node trust evaluation using the received sensor node-sensor node trust evaluation results. Finally, the cluster head node performs a cluster head-cluster head trust evaluation of other cluster head nodes. On the basis of the trust evaluation value, attacks including the black-hole attack are detected.

Misra et al. [7] proposed a scheme named BAMBi that prevents black-hole attacks in wireless sensor networks using a gateway-assisted trust evaluation technique. BAMBi utilizes four BSs to make all sensor nodes send data to each of them. This allows the BSs to aggregate data and detect the attack areas where the black-hole attacks occurred. Moreover, each sensor node periodically sends some type of beacon message called PINFO to the four BSs. PINFO is a message that contains information about the parent node in the routing tree when the base station is set to root on each sensor node. Each base station uses PINFO and the previously detected attack areas to calculate the routing paths and find malicious nodes that perform black-hole attacks.

### 2.2. Gateway-Independent Trust Evaluation

Asai et al. [8] proposed a trust model named TMUMO that prevents gray-hole attacks through the use of nodes utilizing a gateway-independent trust evaluation technique. In this model, the trust value and usage value are derived from the packet forwarding ratio and the amount of usage by the node for the packet relay, respectively. When the node usage is low, the blacklist threshold is low, and an evaluating node does not get detected as an evaluated node even if the trust value of the latter is low. In contrast, with a higher node usage that leads to a higher blacklist threshold, an evaluating node strictly checks the evaluated nodes.

Jhaveri and Patel [9] proposed a trust model that prevents black-hole and gray-hole attacks by analyzing attack patterns using a gateway-independent trust evaluation technique. The method for detecting the attack patterns is by comparing the sequence number information during the time window, and it determines malicious and normal behavior. The authors presented the details of the three modes of adversary models by launching various types of forwarding misbehavior. Their experimental results showed the integration of a pattern discovery method into the trust model.

Mohanapriya and Krishnamurthi [10] proposed a routing protocol that prevents gray-hole attacks with an intrusion detection system using a gateway-independent trust evaluation technique. It was designed according to the Dynamic Source Routing [13] protocol. Intrusion detection system (IDS) nodes can be set to promiscuous mode when required, to detect malicious behavior in the packets being forwarded by a node. When a malicious node is detected, the IDS node broadcasts a block message, informing all nodes in the network topology to isolate the malicious node from the network.

Marchang and Datta [11] proposed a routing protocol that prevents black-hole and gray-hole attacks using a gateway-independent trust evaluation technique. It was designed according to the ad hoc On-demand Distance Vector (AODV) [14] protocol. In this protocol, each node counts the number of packets that are received/transmitted by neighboring nodes to calculate their trust values. Then, the computed trust values are spread over the network using the route request and route reply messages in AODV, and the routes only consist of nodes with high trust values. Consequently, malicious nodes are gradually isolated from the network.

The common limitations of the aforementioned works are summarized as follows. First, the scheme proposed in [5] has a disadvantage in that the BSs periodically perform a trust evaluation of all nodes to detect an attack whether attack nodes exist or not. This puts a burden on all nodes as they are required to send a control message generated by trust evaluation. Second, the schemes proposed in [6,8,10,11] have a common disadvantage in that every node must inspect all surrounding traffic to perform a trust evaluation. This is not suitable for resource-constrained sensor node environments. Finally, the scheme proposed in [7] has a disadvantage in that four copies of the packet are sent to the BS for trust evaluation. This consumes four times the energy from the point of transmission and forwarding. Moreover, this scheme is not suitable for resource-constrained sensor node environments. Finding an attack is important; however, energy consumption must also be considered. Therefore, we propose a stepwise and hybrid trust evaluation scheme that performs a trust evaluation only when triggered by the local observation step where suspicion is captured within the network.

## 3. Proposed Scheme

### 3.1. A Stepwise and Hybrid Trust Evaluation System

Figure 2 shows the overall structure of the proposed scheme. The proposed scheme is divided into three phases: “observation”, “trust evaluation”, and “inspection result sharing”. First, there is a local observation step in the observation phase to observe the traffic behavior of child nodes by the node itself. The node performs a local evaluation and a local inspection of the child nodes. If the result of the local evaluation is normal, the node returns to the local observation step. Otherwise, the node proceeds to the collaborative observation step with other nodes or with the gateway. On the basis of the results of the collaborative observations, the node moves on to the trust evaluation phase. Depending on the situation in this phase, a gateway-independent or a gateway-assisted trust evaluation is performed. The gateway-independent trust evaluation is performed through a collaborative inspection of the nodes around the suspicious node. In contrast, the gateway-assisted trust evaluation is performed through a physical inspection by a drone node or through a logical inspection by a logical inspection node. We propose a logical and physical inspection method to detect false positive problems. The false positive problem is that an attacker can send false information to the gateway bypassing false results. Note that, in our previous work [12], a process of malicious node observation/inspection is done only in a gateway-independent manner and there is no way for a gateway to judge whether or not an attacking node intentionally reports a normal node’s behavior as if it is abnormal. Such a false positive problem can now be resolved through our novel approach for a gateway-assisted trust evaluation. Finally, if the suspicious node is determined to be a malicious node according to the result of the trust evaluation phase, the evaluating node proceeds to the inspection result sharing phase, which shares information about the malicious node. This sharing phase consists of a local propagation step starting from the nodes or a global propagation step starting from the gateway.

Figure 3 shows a flowchart of the proposed scheme. First, every node performs a local observation step to monitor child nodes. If the result of the local evaluation is normal, the node returns to the local observation step. Otherwise, it proceeds to the collaborative observation step with other nodes or with the gateway. If a node receives information from a parent node that a suspicious node has been identified as a malicious one, it proceeds directly to the inspection result sharing phase without going through the trust evaluation phase.

In the trust evaluation phase, the gateway decides how to perform a trust evaluation. If there are no nodes around the suspicious node that can perform the trust evaluation through a collaborative inspection, the gateway performs a trust evaluation with a physical inspection using a drone node. The drone node that receives the physical inspection duty goes to the area where the suspicious node is located and performs an operation that measures the forwarding ratio for a certain time. When the information obtained through the physical inspection is returned to the gateway, the gateway continues with the trust evaluation. However, if there are several nodes around the suspicious node and if the suspicious node’s trustiness is higher than 0.6, the gateway sends a collaborative inspection response to the nodes around the suspicious node to perform a collaborative inspection. When the nodes around the suspicious node receive a collaborative inspection response, they perform a collaborative inspection that checks whether the suspicious node forwards the packet through overhearing. Trust evaluation is performed using the forwarding ratio calculated by the collaborative inspection. In contrast, if the trustiness of the suspicious node is below 0.6 because of repeated false-positive reports or collaborative observation query, the gateway performs a trust evaluation with logical inspection. It sends a logical inspection duty to the node with the highest trustiness around the suspicious node. The node responsible for the logical inspection performs a trust evaluation with a logical inspection based on overhearing.

If the result of the trust evaluation of a suspicious node is confirmed normal, the evaluating node returns to the local observation step again. Otherwise, if an attack is detected, the node goes to the inspection result sharing phase, which shares information about the malicious node throughout the network. Hereinafter, detailed explanation of the sequence will be given with example pictures.

### 3.2. Observation Phase

#### 3.2.1. Local Observation

In tree-based tactical sensor networks, the tactical data information is repeatedly passed to the parent node to reach the gateway. Therefore, each node would observe the incoming data from its child nodes. A node does not perform an overhearing-based inspection that inspects all traffic around itself, but simply inspects the packet that comes to it. If the number of incoming packets is smaller than the number of packets normally received, a local evaluation is performed to detect anomalies.
(1)$$MA{\left(RP\right)}_{t}=\frac{RP_{t-n}+RP_{t-(n-1)}+\cdots +RP_{\left(t-1\right)}}{n}$$
where $RP_{t}$ is the number of packets received from the child node at *t* seconds and $MA{\left(RP\right)}_{t}$ represents the moving average number of packets received from the child node for *n* seconds before the current time. The reason for using a moving average is that the situation of the tactical network can be changed to calculate the normal period using only historical data. If $RP_{t}$ of the current time at *t* seconds is smaller than $MA{\left(RP\right)}_{t}$, the number of packets received is smaller than the average number of packets. Therefore, if malicious nodes G and J launch an attack (e.g., a black-hole or a gray-hole), the parent node would receive an abnormal pattern of traffic. Then, the parent node would go to the next step called collaborative observation.

Figure 4 shows an example of the local observation step. In the figure, node C is aware of the traffic from nodes F, G, and H. Moreover, node E is aware of the traffic from node J. When the malicious nodes G and J launch an attack (e.g., black-hole and gray-hole), nodes C and E would receive an abnormal pattern of traffic. Then, nodes C and E would enter the next step, i.e., the collaborative observation step. The pseudo code explanation for this phase is shown in Algorithm 1.

#### 3.2.2. Collaborative Observation

Also, Figure 4 shows an example of the collaborative observation step. If a node senses an abnormal pattern of traffic from any of its child nodes, it issues a query message to confirm whether the suspicious node has already been reported as a malicious one. The query message is relayed to the gateway node along the established routing path, as shown in Figure 4. Each node k keeps a list of reported malicious nodes $M_{k}$, and, therefore, nodes A and B in Figure 4 would send a response indicating that nodes G and J have been detected as malicious ones. When the response message is received, nodes C and E will propagate the information about the malicious nodes to their respective child nodes, enabling them to reset their parent nodes.

If no node on the path has information about the suspicious node, the query message reaches the gateway. If the gateway does not have information about the suspicious node either, it goes on to the trust evaluation phase. The pseudo code explanation for this phase is also shown in Algorithm 1.

**Algorithm 1** Algorithm for Observation Phase**Input:** The number of packets received from the child node $RP_{t}$    The moving average number of packets received from the child node $MA{\left(RP\right)}_{t}$    The set of child nodes *v***Output:** Result of observation phase//Local Observation**for** each node *u* calculates $MA{\left(RP\right)}_{t}$ about its child nodes *v*
**do** **for** each node *c* ∈ *v*
**do**  **if** node *c*’s $RP_{t}$ < $MA{\left(RP\right)}_{t}$
**then**   node *c* becomes suspicious node  **else**   node *c* becomes normal node  **end if** **end for**
**end for**
//Collaborative Observationnode *u* send query to parent node *p* about suspicious node *w***while** query reaches to Gateway **do** **if** node *p* has information about node suspicious node *w*
**then**  node *p* sends response to node *u* about suspicious node *w*  **break** **else**  node *p* sends query to its parent node $p^{\prime }$  $p\leftarrow p^{\prime }$ **end if**
**end while**
**if** Gateway has information about node *w*
**then** gateway sends response to node *u* about suspicious node *w*
**else**
 gateway determines trust evaluation about suspicious node *w*
**end if**


### 3.3. Trust Evaluation Phase

In the trust evaluation phase, the gateway decides how to perform trust evaluation for the suspicious node. First, if the suspicious node is in a parent-only evaluation region where there are no nodes around node w to perform the trust evaluation, the gateway decides to perform a gateway-assisted trust evaluation with a physical inspection.

Second, if several nodes around the suspicious node can perform a trust evaluation through a collaborative inspection, the gateway first checks the trustiness *T* of the suspicious node. The reason for checking the trustiness *T* of a suspicious node is to check if there has been a repeating collaborative observation query before. Trustiness *T* is managed by the gateway, and the initial value is 1.0. Like the previous explanation, when a repeated collaborative observation query comes, the gateway manages it by imposing a 10% penalty on the suspicious node. If the node’s trustiness *T* is also satisfied, the gateway decides to perform a gateway-independent trust evaluation by sending a collaborative inspection response.

Finally, if there are several nodes around the suspicious node but the suspicious node’s trustiness *T* is not satisfied, the gateway decides to perform a gateway-assisted trust evaluation with a logical inspection. The gateway determines that the suspicious node’s trustiness *T* is low, the parent of the suspicious node sends false positive reports repeatedly, or the suspicious node performs an intelligent attack. Therefore, the gateway is to observe in more detail by participating in trust evaluation through logical inspection.

In Figure 4, multiple nodes exist around node G to perform a trust evaluation through a collaborative inspection, which causes the gateway to decide to perform a gateway-independent trust evaluation with a collaborative inspection. If the trustiness *T* of node G is low, owing to the repeated collaborative observation reports, the gateway decides to perform a gateway-assisted trust evaluation with a logical inspection. On the other hand, if there are no nodes that can perform a trust evaluation through a collaborative inspection around node J, the gateway decides to perform a gateway-assisted trust evaluation with a physical inspection. The pseudo code explanation for this phase is shown in Algorithm 2.

**Algorithm 2** Algorithm for Trust Evaluation Phase**Input:** The trustiness of the node *k*
$T_{k}$    The node *u* is who sends a query against suspicious node *w***Output:** Determine how to perform trust evaluation for suspicious node *w***if** there are no nodes around suspicious node w to perform the trust evaluation **then** gateway determine Gateway-assisted trust evaluation through physical inspection
**else**
 **if** node *w*’s trustiness $T_{w}$ is higher than 0.6 **then**  gateway determine Gateway-independent trust evaluation through collaborative inspection **else**  gateway determine Gateway-assisted trust evaluation through logical inspection **end if**
**end if**


#### 3.3.1. Gateway-Independent Trust Evaluation

Figure 5 shows an example of the gateway-independent trust evaluation approach. The process of collaborative inspection begins with node C (the node that issued the query) by asking its child nodes to monitor the suspicious node G. Then, the child nodes F and H perform an overhearing-based inspection and collect transmission/reception information to and from the suspicious node G. Finally, they perform a trust evaluation with an adaptive detection threshold.
(2)$$Mali_{k}=\alpha \left(1-T_{k}\right)+\left(1-\alpha \right)U_{k}$$
(3)$$U_{k}=\frac{Forwarding_{k}}{\gamma +Receiving_{G}}$$


Equation (Equation 2) is an adaptive detection threshold calculation based on node trustiness $T_{k}$; it is calculated as the weight ratio of node usage to node trustiness $T_{k}$. In the equation, node usage $U_{k}$ represents the packet reception dependency on node *k* from the gateway perspective. In Equation (Equation 3), node usage $U_{k}$ is the amount of node *k* forwarded data to $Forwarding_{k}$ divided by all of the packets received at the gateway $Receiving_{G}$. Moreover, γ is a constant (γ > 0). Here, the amount of forwarded data and the count of received packet are calculated as the moving average, like in Equation (Equation 1). Therefore, the higher the delivery rate, the higher the node usage for that node. Finally, the malicious node threshold $Mali_{k}$ is computed as α:1−α of the node trustiness $T_{k}$ and the node usage value $U_{k}$. The malicious node is detected by comparing the actual packet forwarding ratio of the suspicious node with the calculated threshold. When node G has a lower forwarding ratio than the threshold, it is detected to be a malicious node and node C proceeds to the inspection result sharing phase. The pseudo code explanation for this trust evaluation is shown in Algorithm 3.

#### 3.3.2. Gateway-Assisted Trust Evaluation

Figure 6 shows an example of the gateway-assisted trust evaluation approach. The gateway selects a node with the highest trustiness *T* among the one-hop nodes of the suspicious node G to send a logical inspection duty. In the figure, the gateway selects node H and sends a logical inspection duty. Node H changes its role from a normal node to a logical inspection node when it receives the duty. When it becomes a logical inspection node, it performs a trust evaluation and an overhearing-based logical inspection of the suspicious node G.

First, the logical inspection node selects five random interval times between 1 and 3 s. The reason for the set random interval variable here is that node H does not know at what interval the gray-hole attack is performed, and therefore the packet forwarding ratio is calculated at various intervals. The packet forwarding ratio is calculated a total of five times at every random interval after the logical inspection node H performs a trust evaluation of node G. If the calculated forwarding ratio values are smaller than the adaptive detection threshold value by more than threefold, node G is identified to be a malicious node and node H proceeds to the inspection result sharing phase. The pseudo code explanation for this trust evaluation is shown in Algorithm 3.

As shown in Figure 6, the gateway sends a drone node to the suspicious node J. The drone node receives a physical inspection response from the gateway and moves to the suspicious node J. Then, the drone performs a physical inspection of the suspicious node J. The drone node does not send any messages to the suspicious node but simply observes and records its packet forwarding. Figure 7 shows an example of packet forwarding according to the time window created on the basis of the information received from the drone node. The first time window in Figure 7 shows that 10 packets are well transmitted, which indicates 100% packet forwarding. In the second time window, 5 out of 10 packets have failed to be forwarded and, thus, only 50% packet forwarding is done.

After a certain period of observation, the drone node returns to the gateway and transmits the recorded packet forwarding information. The gateway performs a trust evaluation of the suspicious node J by analyzing the pattern using the packet forwarding information configured on the basis of the time window. Through this, pattern analysis is performed using the packet forwarding ratio for each time window. Looking at the packet forwarding ratio according to the time window, we can see that the malicious node performing the selective forwarding attacks repeatedly performs normal forwarding and packet-dropping attacks. If an abnormal time window is seen at regular intervals among several time windows, it is considered as a selective forwarding attack. The algorithm simply detects a selective forwarding pattern when more than four out of 10 time windows have a forwarding ratio smaller than the adaptive detection threshold. If the suspicious node J is detected to be a malicious node through a trust evaluation using pattern analysis, the gateway proceeds to the inspection result sharing phase. The pseudo code explanation for this trust evaluation is shown in Algorithm 3.

### 3.4. Inspection Result Sharing Phase

Figure 8 shows an example of the inspection result sharing phase. There are two ways to share the results of a trust evaluation: local sharing and global sharing. Local sharing starts to propagate them from the node that performed the trust evaluation, whereas global sharing starts from the gateway. As an example of local sharing, when the malicious node G is successfully detected by the collaborative inspection, node C will begin the process of resetting the routing paths for the child nodes. Concurrently, information about the malicious node G is propagated starting from node C to the gateway along the established routing path, which minimizes the control overhead while allowing the nodes near the attack area to know about the malicious node. In the case of logical inspection, node H informs node C about the malicious node G and propagates the same way as described above.

An example of global sharing is the gateway along the established routing path spreading the information about the malicious node J throughout the network.

**Algorithm 3** Algorithm for Gateway-independent and Gateway-assisted Trust Evaluation**Input:** The trustiness of the node *k*
$T_{k}$    The node *u* is who sends a query against suspicious node *w*    The forwarding ratio of the node *k*$For_{k}$    The malicious node threshold of suspicious node *w*$Mali_{w}$    The set of child nodes *v*    The logical inspection node *l***Output:** Whether the suspicious node is malicious or normal//Collaborative inspectiongateway sends collaborative inspection response to node *u*node *u* relay response to child nodes *v* except suspicious node *w***for** node *c* ∈ *v*
**do** node *c* collects transmission/reception information node *c* reports to node *u*
**end for**
node *u* calculates $For_{w}$ about suspicious node w**if**$For_{w}$ < $Mali_{w}$
**then** suspicious node *w* becomes malicious node
**else**
 suspicious node *w* becomes normal node
**end if**
//Logical inspectiongateway selects node *l* with the highest trustiness *T* among 1-hop nodes of suspicious node *w*gateway sends logical inspection response to node *l*
count←0
**for***i* = 1 to 5 **do** node *l* selects random interval time $t_{i}$ node *l* calculates $For_{w}$ in $t_{i}$ seconds about suspicious node *w* **if**
$For_{w}$ < $Mali_{w}$
**then**  count++ **end if**
**end for**
**if**count > 2 **then** suspicious node *w* becomes malicious node
**else**
 suspicious node *w* becomes normal node
**end if**
//Physical inspectiongateway sends physical inspection response to drone node
count←0
drone node goes to suspicious node *w* areadrone node collects transmission/reception information about suspicious node *w***for***i* = 1 to 10 **do** gateway calculates $For_{w}$ in *t* seconds about suspicious node *w* **if**
$For_{w}$ < $Mali_{w}$
**then**  count++ **end if**
**end for**
**if**count > 4 **then** suspicious node *w* becomes malicious node
**else**
 suspicious node *w* becomes normal node
**end if**


## 4. Performance Evaluation

### 4.1. Simulation Environment

In this section, we discuss the result of the performance evaluation of the proposed scheme while using the OPNET simulator. This simulator was chosen as the main tool for simulating our proposed scheme because its implementation stack is very similar to a real-world network stack. The simulation was performed with 30 nodes that were placed in a tree topology in a 6000 m × 6000 m area. There was a fixed gateway node and 29 other sensor nodes. We assume that the drone node in this simulator is a micro tactical drone named Black Hornet [15]. This drone is developed by the British army and used in Afghanistan. The drone’s specs are ~17 cm in size, 20 km/h at top speed, and can last up to 25 min. The communication range has a wireless communication radius of 2 km. We experimented by constructing a tree-structured route with 0–4 malicious nodes that performed different attacks to verify the performance of detecting several attacks (e.g., gray-hole, false positive, and smart gray-hole) at varying rates. Among the gray-hole attacks, an attack that continuously discards packets at a constant rate is called a sequence number-based gray-hole attack and one that interrupts and re-executes an attack depending on a situation is called a smart gray-hole attack [16]. The traffic types of the tactical sensor data were set in consideration of the type, size, and generation cycle of the chemical, biological, and radiological sensors, and of the surveillance videos [17,18]. The packet generation rate was set to 110 Kbps. The energy consumption and packet delivery ratio of the proposed scheme were compared to those of CENTERA [5], which supports only a gateway-assisted trust evaluation system and our previous work [12]. As described in related works, CENTERA is a scheme in which the base station calculates packet forwarding for each node using only the reports of its child nodes and uses it to give an isolated time that meets the corresponding threshold. The simulation environment parameters are listed in Table 1.

### 4.2. Simulation Results

In this section, we discuss the simulation and analysis results. We focus more on the packet delivery ratio and energy consumption for the simulation result, as our goal is to reduce the energy consumption and to detect malicious nodes by proposing a stepwise and hybrid trust evaluation scheme.

Figure 9a shows the packet delivery ratio in a gray-hole attack depending on the attack percentage. Gray-hole attacks were executed by two malicious nodes, with 10–90% ratio of dropped packets. Moreover, the malicious nodes started the gray-hole attacks from 10 s. First of all, the proposed scheme is shown to maintain a high packet delivery ratio of over 95%. On the other hand, CENTERA also shows that it maintains a relatively high packet delivery ratio when the attack percentage is over 40%. However, at 10–30% attack rates, the packet delivery ratio of CENTERA decreases. For this reason, CENTERA discriminates against various malicious node thresholds and uses the exclusion time set in each section to exclude the malicious node before rejoining the network. CENTERA divides the seven malicious thresholds from 10% to 70% in 10% intervals. Intervals with lower malicious thresholds have longer isolated times. As a result, a malicious node with a lower packet forwarding ratio has a longer isolated time. Thus, CENTERA did not detect attacks with 10–30% attack rates, which affected the overall packet delivery ratio. For similar reasons, our previous work also failed to detect gray-hole attacks below 40% due to a 60% fixed malicious threshold. However, our proposed scheme could detect malicious nodes owing to the adaptive detection threshold by the reliability and usage of nodes. Another reason is that the gateway participated in the trust evaluation through a logical inspection only for unreliable suspicious nodes. Moreover, the proposed scheme showed that the packet delivery ratio was 1–5% higher than that of CENTERA at an attack rate of 40–90%. The reason is that, in CENTERA, when the exclusion time of the malicious node is over, it reenters the network and resumes the trust evaluation. Thus, the cycle will take longer, but the attack and discrimination will occur repeatedly, affecting the overall packet delivery ratio. In this paper, we experimented by blocking the detected malicious nodes by emphasizing the security importance of tactical data. However, it can be involved depending on the operation of the user. In conclusion, the proposed scheme is more effective at detecting gray-hole attacks than CENTERA. Moreover, the proposed scheme allows for more detailed gray-hole observations than those of CENTERA.

Figure 9b shows the energy consumption in gray-hole attacks depending on the attack percentage. Energy consumption was calculated as the amount of power used in the transmission (Tx) and receipt (Rx) of the control message delivered for trust evaluation. The Tx energy of the control message used in the performance evaluation of CENTERA was 96 (µJ) and the Rx energy was 107.2 (µJ). Therefore, the same values were applied to the consumed energy of the proposed scheme. First of all, CENTERA showed a higher energy consumption of 320–350 (µW) than that of the proposed scheme and previous scheme. The reason is that all sensor nodes of CENTERA report control messages periodically, regardless of whether they are attack nodes. On the basis of this result, the gateway performed a trust evaluation to detect malicious nodes and to isolate them from the topology exclusion time. In comparison, the proposed scheme consumed significantly lower energy because, in this scheme, trust evaluation is triggered by the local observation step where the suspicion is captured within the network. Moreover, in this scheme, the consumed energy decreases as the attack percentage increases. The reason is that the higher the attack percentage, the more obvious the attack pattern on the network. Therefore, trust evaluation is quicker to identify attacks, resulting in fewer control messages. However, in 50–90%, the previous scheme and proposed scheme show similar energy consumption, whereas in 10–40%, the previous scheme shows higher energy consumption than the proposed scheme. This is because the previous scheme selected a malicious threshold value of 60% and performed repeated trust evaluation for gray-hole attacks below 40%. In conclusion, the proposed scheme is much more energy efficient than CENTERA. Moreover, as the simulation time and the number of nodes increase, so will the difference in energy consumption between CENTERA and the proposed scheme.

Figure 10 shows an average packet delivery ratio and energy consumption, respectively, depending on the number of malicious nodes when a gray-hole attack occurs. The gray-hole attack here is assumed to be executed by malicious nodes, with the 40% ratio of dropping packets. As shown in the figures, the packet delivery ratio is consistently decreasing with the increase of malicious nodes but not much different for both schemes’ performance (even though the proposed scheme looks performing slightly better than CENTERA in Figure 10a). More interesting results are presented in Figure 10b as it proves the fact that our scheme performs significantly better from the energy efficiency perspective and the impact of varying number of malicious nodes against its performance seem to be minor. This is because trust evaluation is triggered by the local observation step where the suspicion is captured within the network. Whereas, CENTERA shows much higher energy consumption of 320–350 (µW) than that of the proposed scheme. The reason is that all sensor nodes in CENTERA are required to report control messages periodically, regardless of whether they are attack nodes, causing the significant energy consumption.

Figure 11a shows the packet delivery ratio in smart gray-hole attacks. A smart gray-hole attack is a type of intelligence attack that detects the characteristics of the protocol used and maintains a certain level of reliability. In CENTERA, each time a node is detected as a malicious node, the reliability of node increases. On the other hand, when the behavior of a node is good, the reliability of node approaches 1. In CENTERA, each time a node is detected as a malicious node, the trust value of “banNum” increases. On the other hand, when the behavior of a node is good, the trust level of “banNum” approaches 1. “banNum” means the cumulative number of warnings for each node used in CENTERA. This value increases by 1 every time it is detected as an attacking node and decreases by 1 whenever normal behavior is observed. Therefore, if a smart gray-hole attack is detected as a malicious node while performing an attack, it is excluded from the network for a certain time and recovers reliability of node by doing a good forwarding pattern when it joins the network again. After that, the attack is executed again and the action is repeated to maintain a certain trust and to continuously attack. In the proposed scheme, when a node detects a suspicious node through the local observation step, the malicious node knows that there is information about itself in the collaborative observation response. Therefore, if it performs a good forwarding pattern for a certain period time, the attack will not be detected as a result of the trust evaluation and the process returns to the local observation step. By repeating this behavior, the malicious node can maintain a certain level of reliability and attack constantly. Therefore, in Figure 11a, CENTERA shows that the packet delivery ratio rises and falls over time. However, the proposed scheme had a lower packet delivery ratio than that of CENTERA before 50 s, which rose to 100% of the packet delivery ratio after 50 s. This is because the proposed scheme does not perform a gateway-independent trust evaluation through a collaborative inspection when the reliability of the suspicious node is below a certain level. In that situation, the proposed scheme performs a gateway-assisted trust evaluation through a logical inspection. Therefore, the malicious node cannot know that it is being observed and attacked even if a logical inspection is being performed. Therefore, the proposed scheme showed a lower packet delivery ratio than that of CENTERA because the attack continued before 50 s. On the basis of the results of the logical inspection, the gateway succeeded in detecting the malicious node and isolated it from the network to achieve a packet delivery ratio of 100%.

Figure 11b shows the energy consumption in a smart gray-hole attack. We used the same Tx and Rx energy consumption values as those of CENTERA, as described in Figure 9b. First, both CENTERA and the proposed scheme showed an increase in energy consumption before 50 s. However, after 50 s, only CENTERA continued to increase and the proposed scheme did not consume any more energy. The reason is the same as that explained in Figure 11a. All sensor nodes of CENTERA report control messages to the gateway periodically for trust evaluation, regardless of whether there are attack nodes. In comparison, the trust evaluation of the proposed scheme is triggered by the local observation step where suspicion is captured within the network. However, the proposed scheme did not perform a gateway-independent trust evaluation through a collaborative inspection when the reliability of the suspicious node was below a certain level, as shown by the results in Figure 11a. In that situation, the proposed scheme performed a gateway-assisted trust evaluation through a logical inspection. Therefore, the gateway succeeded in detecting the smart gray-hole attack around 50 s. After 50 s, energy was only consumed to propagate the information about the malicious node to the gateway. This difference also increased with the simulation time and the number of nodes, as shown by the results in Figure 9b. In conclusion, the proposed scheme is much more energy efficient in trust evaluation than CENTERA and it is also capable of intelligent attack detection based on reliability.

Figure 12 shows the cumulative distribution function of end-to-end delay in false positive attacks. In our previous scheme [12], any attacking node may trigger a collaborative inspection by performing local observation on its subordinate nodes. We assume that the attacking malicious node intentionally reports a gateway about its child node as an abnormal one and blocks this node from the network. Then, the tree-based sensor network topology towards the gateway becomes changed to find out some bypass path, resulting in a longer delay for data transfer (~13.5% longer end-to-end delay in the figure) compared to the proposed scheme where such a false-positive problem can be detected via a gateway-assisted evaluation phase—a physical inspection using a special mission-oriented drone in this scenario.

## 5. Conclusions

Owing to the limited battery capacity of tactical sensor nodes in tactical wireless sensor networks, trust evaluation must be provided in an energy-efficient way. Among many security issues, this paper focuses on several packet-dropping attacks (namely, black-hole, gray-hole, and smart Gray-hole) that could be dangerous in tactical environments. The results of simulation experiments show that the proposed scheme successfully detects the malicious nodes that performed two types of packet-dropping attacks (gray-hole and smart gray-hole) with various attack ratios, resulting in a higher packet delivery ratio and lower energy consumption than those of CENTERA. We also avoid false-positive problems compared to previous work. This indicates that, although we mainly considered unmanned ground sensors in this study, the proposed scheme can also be applied to various communication systems with similar traffic characteristics such as smart grid networks and energy-efficient sensor network communication. In a future work, we will try to improve the trust evaluation scheme further by using a machine learning method to provide situation awareness penalties and rewards.

## Figures and Tables

**Figure 1 sensors-20-01108-f001:**
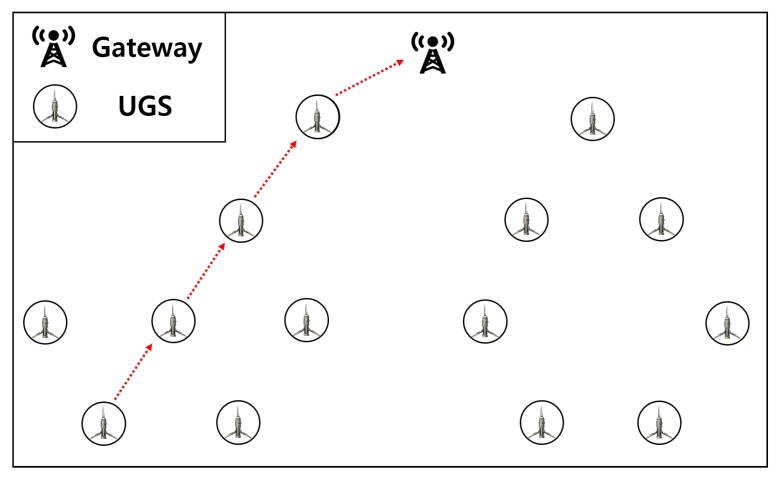
Illustration of a tree-based tactical wireless sensor network.

**Figure 2 sensors-20-01108-f002:**
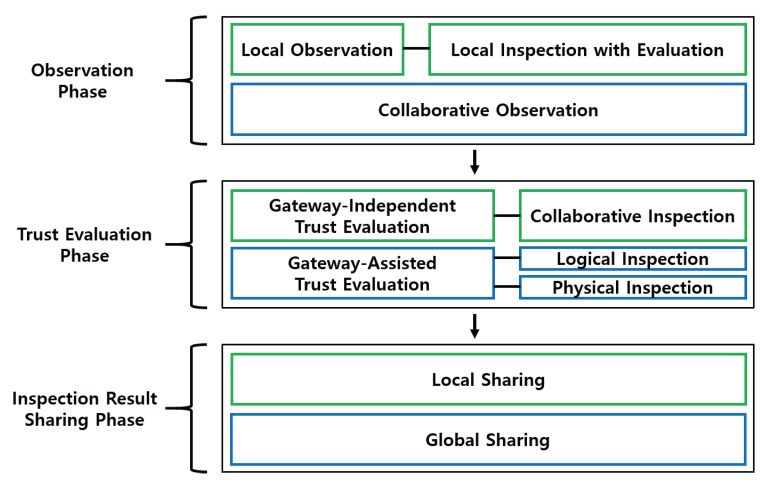
Overall structure of the proposed scheme.

**Figure 3 sensors-20-01108-f003:**
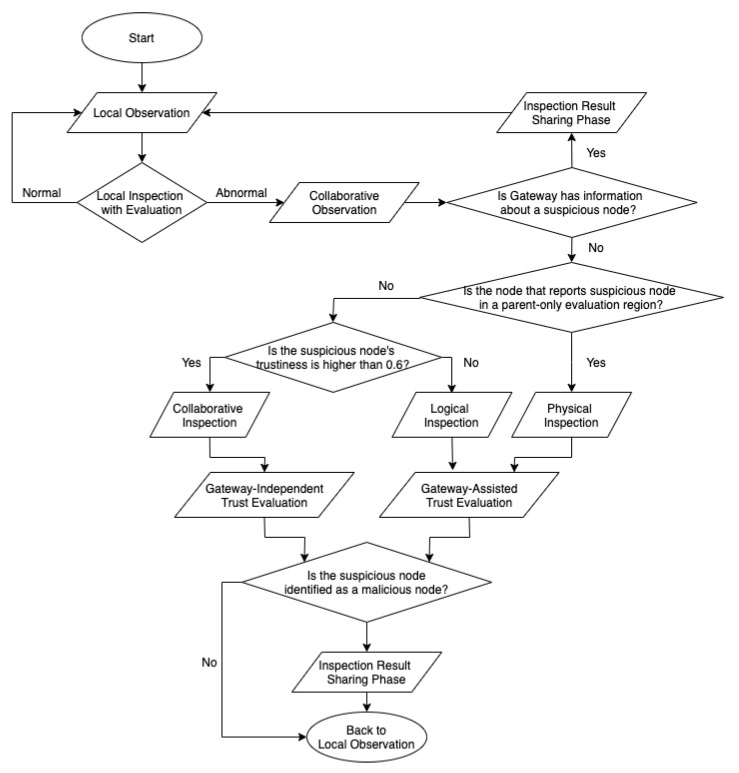
Flowchart of our trust evaluation system.

**Figure 4 sensors-20-01108-f004:**
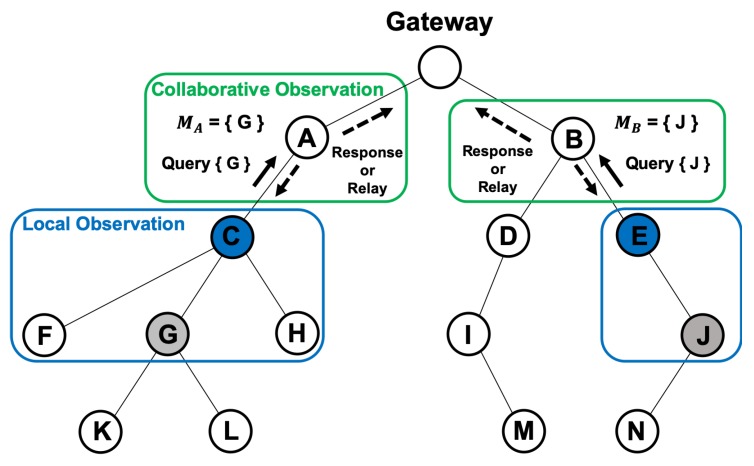
Example of the observation phase consisting of 2 steps.

**Figure 5 sensors-20-01108-f005:**
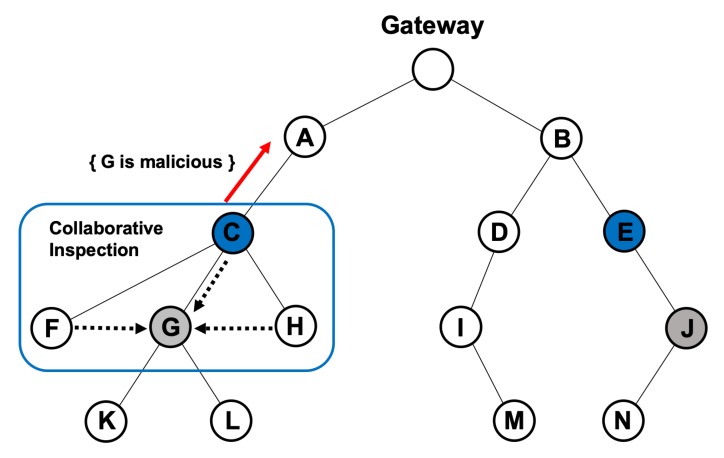
Example of the trust evaluation phase with the collaborative inspection step.

**Figure 6 sensors-20-01108-f006:**
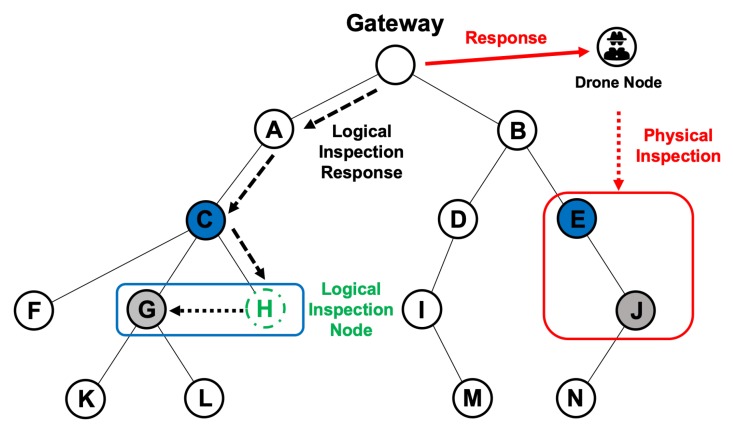
Example of the Trust evaluation phase with logical and physical inspection.

**Figure 7 sensors-20-01108-f007:**
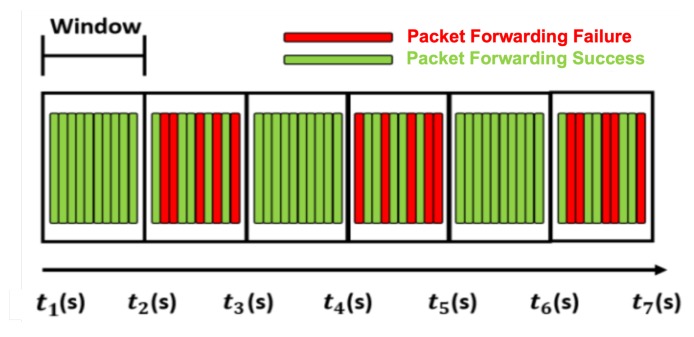
Packet forwarding according to the time window.

**Figure 8 sensors-20-01108-f008:**
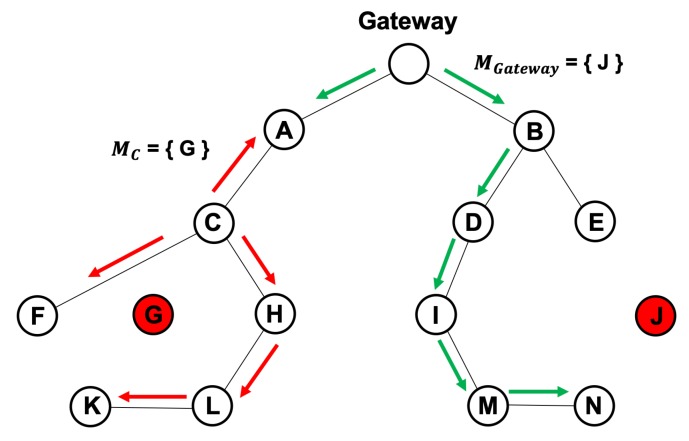
Example of the inspection result sharing phase.

**Figure 9 sensors-20-01108-f009:**
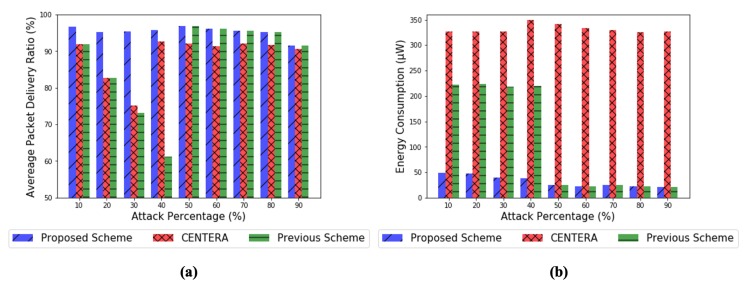
(**a**) Packet delivery ratio and (**b**) energy consumption in the case of gray-hole attack by varying attack rates.

**Figure 10 sensors-20-01108-f010:**
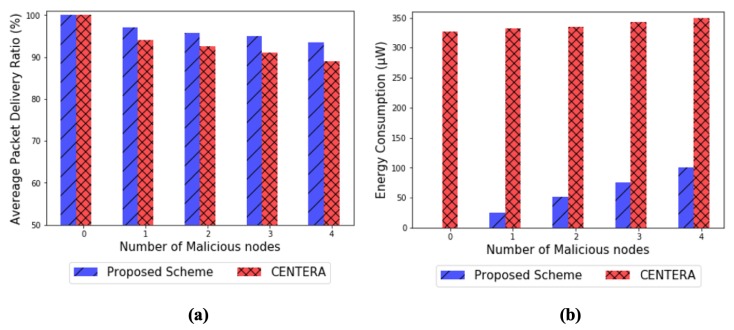
(**a**) Packet delivery ratio and (**b**) energy consumption in the case of Gray-hole attack by varying number of malicious nodes.

**Figure 11 sensors-20-01108-f011:**
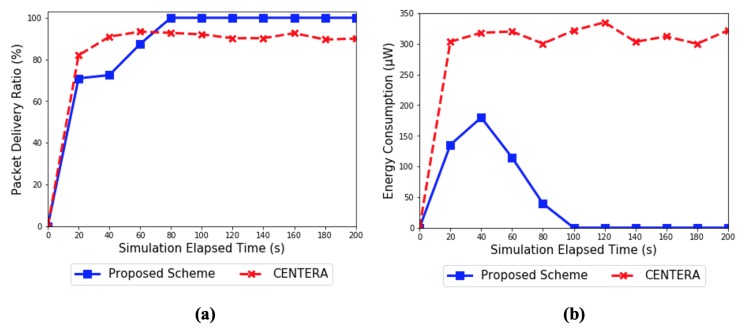
(**a**) Packet delivery ratio and (**b**) energy consumption in the case of smart gray-hole attack.

**Figure 12 sensors-20-01108-f012:**
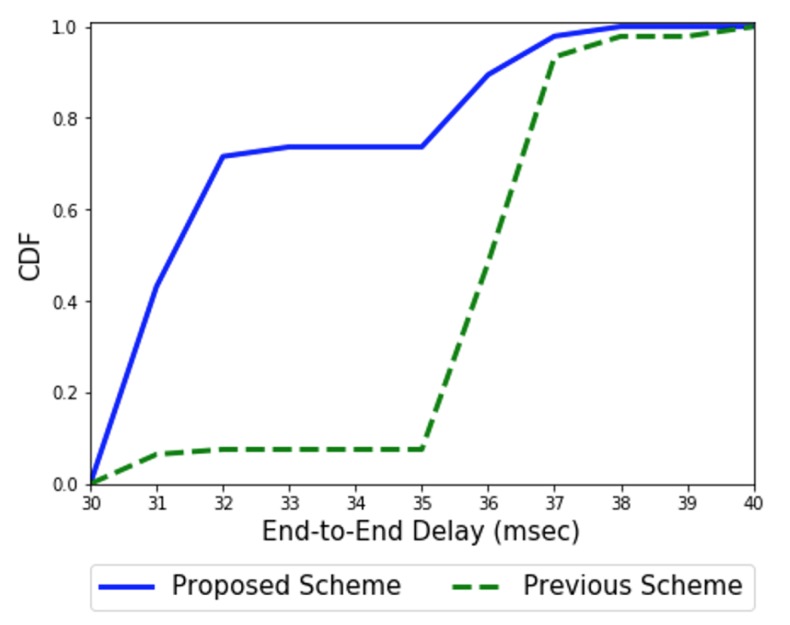
End-to-end delay in the false positive attack.

**Table 1 sensors-20-01108-t001:** Simulation parameters.

Parameters	Values
Simulator	OPNET 18.0
Network size	6000 × 6000 m
Simulation time	200 s
Number of nodes	30 (including 1 gateway)
Number of malicious nodes	0-4
Attack model	Gray-hole or smart gray-hole
Gray-hole attack ratio	10–90%
Traffic type	Lighting sensor (100 bytes)
	Chemical and biological sensors (120 bytes)
	Video surveillance H.264 (500 bytes)
Packet generation rate	110 Kbps
Threshold weight (α)	0.5
Time Window (*t*)	2 s

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
