# Peer review of "A Stepwise and Hybrid Trust Evaluation Scheme for Tactical Wireless Sensor Networksâ€"

_sensors, 2020, doi:10.3390/s20041108_

Round 1
Reviewer 1 Report
This paper proposes a trust-based evaluation scheme to locate malicious Unattended Ground Sensors (UGSs) that can perform packet-dropping attacks over their network of UGSs as per my understanding. The proposed schemes include local (or individual) and collaborative trust-based evaluation in making decision about the suspicious level of those sensors showing abnormal behaviors. Although the paper provides flowcharts, graphs, and detailed explanations in describing the proposed schemes, it is not clear how the proposed schemes depend on whether the telecommunication network is wireless, tactical, mobile, or stationary. That is, the interaction between the sensor network of UGSs and the wireless tactical network of communications is not clear. It seems that the proposed trust-based and other schemes do not have anything to do with the communications and networking of wireless tactical networks. I suggest that the authors address this issue and clarify if there is any interaction. If there is no such an interaction between the sensor network of UGSs and the wireless tactical network of communications, the title of the paper looks misleading.
Reviewer 2 Report
In this paper, the authors are proposing a methodology to detect malicious nodes in a wireless sensor network. The authors have also validated their methodology using OPNET simulations and with comparisons of CENTERA approach. In general, the paper is well written and with good organizational structure. However, there are some comments below which need to be addressed before its acceptance. Therefore, I recommend major revision.
On page 2 line 62, the authors mention about “remaining static”. What does it mean? Some of the parameters inside the equations are missing for table 1. For example what is \alpha of Eq. (2)? More thorough link between the notation used and numerical values should be established Give pseudo-code of the proposed approach as a summary format. In its current form, the approach is written as a text version a better presentation is needed using all the notations and equations in a pseudo code format. On page 10 line 276, why did you select 4 out of 10 time windows? CENTERA is not references as shown on page 11 line 306. More details about the CENTERA approach is necessary inside the text as in its current form no details are provided about its characteristics and differences with the proposed scheme. In abstract on line 14, “and the results showed…” On page 9 line 254, it should be “here is that node H” rather than “node G” What is “banNum” on page 13 line 352? On page 14, line 368, “and attacked even if” On page 14, line 377, “CENTERA report control messages” On page 14 line 388, “capable of intelligent attack…”
Reviewer 3 Report
This paper presents an extension to a previous work:
https://doi.org/10.1109/ICTC.2018.8539353
The fact that this is an extension is correctly mentioned by the authors.
One common issue with extensions is that often papers are not self-contained,
this is not the case.
However the author do not clearly identify the additional contributions.
The original work presented a novel approach to malicious node detection in a wireless sensor network. The goal was to reduce the resource usage by exploiting the tree network structure in order to operate a monodirectional (network edge towards base station) hearing and evaluation.
One limitation of such approach is the lack of protection in case of bidirectional network (if the network comprises both sensors and actuators, it may actually be a problem).
The abnormality detection process was performed in two phases, at first by a single device, then confirmed collaboratively by siblings of the suspicious node by overhearing (listening to transmissions of which they are not the intended recipient).
This work extends the previous one by adding a gateway-assisted trust evaluation.
It comes in two flavors, a "logical" inspection and a "physical" inspection.
The former looks like a reduced version of the collective analysis performed in the previous phase.
In this proposal, in fact, the node with highest trust among those within a 1-hop (physical) distance from the suspicious one is appointed as "logical inspection node". Its goal is now performing overhearing, detecting maliciousness, and send results back to the base station. It's unclear why this is needed, which are the cases that cannot get detected by the original collaborative inspection. It looks to me very much forced.
The flowchart states that this method gets chosen when "the gateway is joining the trust evaluation", but no clear indication is provided to what reasonable policy should a gateway apply to decide whether to join the trust evaluation.
The physical inspection is performed when "the suspicious node is in an isolated area". It sounds insufficient to me. The real key seems to me that sibling nodes are relevant to perform a trust evaluation, however I see no reason why the logical inspection could not get performed by the suspicious node's parent directly. As such, I would argue that the most possible isolated nodes are either network leaves or disconnected from the rest of the network. As such, I can see no case in which the physical inspection is actually required, unless used for confirmation in case the parent is the only evaluator.
The physical evaluation requires the use of a mobile device (the authors suggest a drone, but I'd say that anything equipped with hearing capabilities and mobility could work), which enters comm radius, listens to the channel, records, and after a not better defined "certain period" of time flies back to the gateway and transmits the recorded information, which is analyzed to detect possible maliciousness.
This approach has a number of drawbacks in the field. Attackers may target isolated nodes by purpose to attract drones there and tear them down, or, worse, prepare poisonous packets to be reliably brought back to the gateway. As currently described, the solution looks to me as a harbinger of security weaknesses rather than a solution to potential issues.
It could probably make more sense in case the target were safety rather than security: in case of no active attacker, sending a device to inspect unexpected malfunctions could be reasonable.
Overall, I'm not convinced of the soundness and appropriateness of the proposed extension.
I also have some remarks for the evaluation.
First, results are obtained by simulation. This is totally ok to me. However, they are not reproducible. No code, no configuration, no instructions are provided. I can see no obvious reason for this, as there is no real world defense data, nor personal data of any kind. I believe results obtained by simulation should always be shared for the sake of reproducibility.
The presented results are a bit superficial.
* The drone intervention does not seem to enter the evaluation at all. I would have been very curious to understand how big is its impact overall.
* The sentence "CENTERA discriminates... network" is very unclear, and makes it impossible to understand which are the "various thresholds" set at 70%. It's impossible to understand whether the baseline was misconfigured.
* The only studied case features two malicious nodes. I believe the authors need to measure what happens with a varying number of malicious nodes, including no malicious node: in Figure 10, in fact, I'd have expected both approaches to perform with a close-to-perfection delivery rate with no attack.
* The criteria used to decide which two nodes are malicious are not discussed. I believe the authors should test with malicious nodes in diverse locations along the network and measure the impact.
* Energy consumption is provided in terms of energy (joules), but it is not clear on which time scale. It is more relevant to measure consumption in terms of its time derivative power (watts). It won't change the result, but it will provide a much clearer insight. In fact, given the lower power drain by the proposed approach, the longer the experiment the higher the overall energy use. Using power, the evaluation will get independent (besides the startup transition) from the execution time.
* The energy evaluation does not take into account the drone (I suppose, as it'd need at least a hardware description to guess the consumption). It's reasonable to do so, but please make it explicit.
* Some of the differences with CENTERA arise because of it allowing nodes to re-join the network after some time. This can is possibly bad for security, but very reasonable for safety -- namely, to achieve higher resilience, in case the erratic behavior is due to temporary malfunctions rather than maliciousness.
I strongly suggest the authors to re-submit a substantially improved paper, where they at least:
* bring evidence for the practical and economical feasibility of using drones as node inspection devices in tactical networks
* Evaluate the algorithm phases singularly in several different conditions, in such a way to understand clearly which of the strategy to use in which situation
* Evaluate the hybrid strategy along with its single components and CENTERA
* Improve the evaluation, and make the experiments available and reproducible.
Once these elements are in place, there are a number of presentation changes that could help further improve, first of all charts quality. The current ones look like those generated with non-specific software (like spreadsheets), I suggest the authors to look into python/matplotlib, python/seaborn, R, or GNUPlot for better charting.
Round 2
Reviewer 2 Report
The authors have responded to my previous comments.